



# Sensitivity experiments with ICON-LAM to test probable explanations for higher ice crystal number over Arctic sea ice vs. ocean

Iris Papakonstantinou-Presvelou[1] and Johannes Quaas[1]

[1]Leipzig Institute of Meteorology, Leipzig University, Leipzig, Germany

**Correspondence:** Iris Papakonstantinou-Presvelou (i.presvelou@uni-leipzig.de)

**Abstract.** Arctic warming is causing the sea ice to retreat, exposing larger areas of open ocean. This shift is expected to alter aerosol emissions, potentially influencing cloud microphysics and ratiative properties. Recent studies show that ocean-derived ice nucleating particles are highly efficient in ice nucleation at relatively warm temperatures. However, our previous study, a 10-year analysis of ice crystal number concentration from DARDAR-Nice satellite retrievals revealed higher ice numbers over
sea ice compared to open ocean (Papakonstantinou-Presvelou et al., 2022). In the current study we explore potential causal explanations behind this contrast. We perform kilometer scale resolution simulations with the ICON-LAM atmospheric model for a 3-day period in March 2019. We validate the satellite retrievals with recent aircraft in situ observations collected during the AFLUX campaign, finding general agreement in the observed contrast of ice crystal numbers between sea ice and open ocean. Sensitivity experiments investigate three potential mechanisms; enhanced INP concentrations over sea ice, blowing
snow particles and secondary ice production (SIP). Our results suggest that both INPs over sea ice and blowing snow particles can explain the observed difference below $-10°$C, while secondary ice production cannot account for the difference above $-10°$C. We thus conclude that as sea ice retreats in a warming Arctic, the two former processes become less relevant, leading to lower particle concentrations in a future climate.

## 1  Introduction

The Arctic region is particularly susceptible to climate change. The Arctic is warming at a faster rate than the rest of the globe (double or more in Wendisch et al., 2017; Rantanen et al., 2022) and that amplified warming – referred to as Arctic Amplification – is the result of many intertwined processes and feedbacks (Serreze and Barry, 2011; Wendisch et al., 2023). One aspect of the multifaceted puzzle is related to clouds and their interactions with aerosols (Wendisch et al., 2019). In a warming Arctic where the sea ice retreats, there is an expanding uncovering of open ocean. There is evidence that the sea

surface layer and/or the melting sea ice is a source of efficient biogenic particles that can act as ice nuclei (Wilson et al., 2015; DeMott et al., 2016; Zeppenfeld et al., 2019). It was proposed that under global climate change, this loop between the biogeochemistry and clouds is likely to play an important role for the Arctic climate system, by impacting the cloud microphysics and cloud radiative characteristics (Wilson et al., 2015; Schmale et al., 2021; Murray et al., 2021). Thus, the need





to understand the processes and interactions involved in the rapidly changing Arctic climate system is eminent now more than
ever.

The most prominent type of clouds observed in the Arctic is mixed-phase, where ice and liquid coexist in the same volume
and typically consist of a liquid layer on the top and falling ice crystals below (Shupe et al., 2006; Gayet et al., 2009; de Boer
et al., 2009). Such clouds warm the surface most of the year except in summer (Intrieri et al., 2002; Shupe and Intrieri, 2004),
and compared to their lower latitude counterparts are relatively long-lived and persist for a few days (Morrison et al., 2012,
and references therein), thereby enhancing their warming effect. In contrast to pure liquid clouds, where ice is involved, the
situation becomes quite complex. Ice crystals exhibit different non-spherical shapes and span larger sizes compared to droplets,
and as a result they have different radiative properties. Additionally, they exchange mass with liquid droplets, depending on the
thermodynamical conditions. The evolution of mixed-phase clouds is mainly governed by the Wegener-Bergeron-Findeisen
(WBF) mechanism, where ice grows at the expense of liquid droplets, but ice crystals and liquid droplets can also grow
simultaneously or evaporate (Korolev, 2007; Fan et al., 2011). Due to the complexity surrounding Arctic mixed-phase clouds
and their importance for the climate system, a better understanding of relevant processes is needed to reduce uncertainty in
model simulations (Tan et al., 2023). If there is a special source of ice nucleating particles (INPs) over open ocean, the sea ice
retreat implies more frequent mixed-phase clouds, replacing supercooled liquid water clouds, and thus an enhancement of their
peculiar radiative effects. In a previous study, we have analyzed satellite retrievals of the ice crystal number concentration over
sea ice and ocean at a large scale within the Arctic (Papakonstantinou-Presvelou et al., 2022). In contrast to the expectation,
we found larger ice number concentrations over sea ice than over open ocean. This motivated the present study that employs
an atmospheric model to investigate possible causes for this result.

Model simulations spanning different spatial and time scales have been performed in the past. Large eddy simulations (LES)
of Arctic clouds with different models showed that the representation of a correct partitioning between ice and liquid is a
determinant for mixed-phase clouds longevity (Ovchinnikov et al., 2014; Solomon et al., 2015). According to Morrison et al.
(2011) two main cloud states were recorded during an intercomparison of six different cloud-resolving models; persistent
mixed-phase clouds or all-ice state clouds and that depending on the ice nuclei concentrations, a rapid transition from the first
to the other can take place, suggesting a need for better representation of ice nucleation in models. LES simulations with the
COSMO model in the Arctic also investigated clouds over different surface conditions, namely over sea ice and ocean and found
that clouds over sea ice tend to be homogeneous, thin stratus, whereas over the ocean are mostly stratocumulus (Eirund et al.,
2019). In addition, many recent studies performed simulations with the German Weather Service's operational ICOsahedral
Non-hydrostatic (ICON) model to study Arctic clouds (Ruiz-Donoso et al., 2020; Costa-Surós et al., 2020; Kretzschmar et al.,
2020; Schemann and Ebell, 2020; Kiszler et al., 2023; Possner et al., 2024) and related processes in the Arctic (moisture
intrusions in Bresson et al., 2022; Kirbus et al., 2023). In Ruiz-Donoso et al. (2020) they compared simulated quantities of ice
water content from ICON-LES with observations for a warm air advection case in the Arctic and found that the cloud ice in
the model was lower, but the ice particles were mainly present at the cloud top. In Kiszler et al. (2023) they also performed
ICON-LES simulations in the Arctic and assessed the model's ability to reproduce observations, concluding that the model
is in good agreement, i.e low bias in water vapor, but it produces an unrealistic number of ice clouds. Finally, Schemann and



Ebell (2020) linked the low simulated radar reflectivities to presumably overestimated cloud condensation nuclei (CCN) and
ice nucleating particles (INP).

Cloud formation is inherently connected to the ambient aerosol types and concentrations. In particular, INPs are a crucial
part of heterogeneous ice nucleation in mixed-phase clouds in temperatures between $0°$ and approximately $-38°$C. Many
observational studies in the Arctic have measured biogenic aerosols that are able to nucleate ice in relatively warm temperatures
above $-15°$C (Šantl-Temkiv et al., 2019; Pereira Freitas et al., 2023) and some attribute their existence to a marine source (Irish
et al., 2019; Wex et al., 2019; Welti et al., 2020; Hartmann et al., 2020, 2021) and more specifically to biogeochemical activity
in the ocean (Dall'Osto et al., 2017; McCluskey et al., 2017; Creamean et al., 2019). One main mechanism proposed for this
process is related to sea spray particles that contain highly efficient INPs, which through wave and bubble break-up can become
airborne (Quinn et al., 2015; Wilson et al., 2015; DeMott et al., 2016). Other studies in the polar regions find such biogenic INPs
within the sea ice and relate it to melting processes, melt ponds and the marginal sea ice zone (Dall'Osto et al., 2017, 2022;
Creamean et al., 2022; Zeppenfeld et al., 2019, 2023) or to thawing permafrost (Creamean et al., 2020). Recent studies in the
Arctic also claim that measured biological particles originate from terrestrial sources (Perring et al., 2023; Pereira Freitas et al.,
2024). Although biogenic particles can be really important in remote locations such as the Arctic (Vergara-Temprado et al.,
2017; Zhao et al., 2021), they are mostly dominant during summer, while dust minerals coming from large-scale transport is the
main contributor to ice nucleation during the other seasons (Si et al., 2019; Yun et al., 2022; Sanchez-Marroquin et al., 2023;
Ansmann et al., 2023). In addition, alternative dust sources have been reported such as melted glaciers in polar environments
(Tobo et al., 2019; Barr et al., 2023) or even dust emissions from the ocean where dust particles were previously suspended
(Cornwell et al., 2020). A recent study by Kawai et al. (2023) shows that dust emitted within the Arctic can be particularly
efficient INP even at higher temperatures, namely between $-5°$ and $-20°$C. Despite the diversity of INP sources in the Arctic,
it is not the only source of uncertainty when it comes to understanding aerosol-cloud interactions in the region. The role of
other contributing mechanisms affecting cloud ice becomes eminent.

There is a consensus in the community that secondary ice production (hereafter referred to as SIP) is a fundamental process
for mixed-phase clouds and is the reason why substantially higher amount of ice crystals has been observed in clouds (Korolev
and Leisner, 2020, and references therein). The most widely established mechanism for SIP is the Hallett-Mossop process
or rime-splintering, where rimed graupel colliding with supercooled droplets produces splinters in a temperature range of
$-3°$C $\leq T \leq -8°$C (Hallett and Mossop, 1974). However, there are more SIP processes that can take place even at lower
temperatures and stages of the cloud development, not all of which are parameterized in models. Recent studies in the Arctic
suggest that mechanisms such as collisional ice break-up (Sotiropoulou et al., 2020) and droplet shattering (Possner et al.,
2024) could be the missing piece in the puzzle of increased ice crystal number concentrations compared to available INPs.
According to Georgakaki et al. (2022) who investigated SIP in alpine mixed-phase clouds using the WRF model, breakup
seems to highly impact ice crystal numbers by an order of three. Sotiropoulou et al. (2024) also tested three SIP missing
processes in NorESM2 model, namely collisional break-up, drop-shattering and sublimation break-up, and found that the ice
model quantities were improved compared to what is produced solely by primary ice. Additionally, Georgakaki and Nenes
(2024) used a machine learning approach to parameterize SIP in mixed-phase clouds in WRF model and they demonstrated





that this approach reproduces the model results well, thereby suggesting that process simplification could prove useful for
GCMs. A correct representation of SIP processes in large-scale models is therefore needed in order to adequately explain the
ice crystal numbers in mixed-phase clouds.

Apart from the SIP, several studies suggest blowing snow particles to be a potential contributor to ice crystal number concentrations and a natural seeding mechanism over snow-covered surfaces (Lloyd et al., 2015; Geerts et al., 2015; Georgakaki et al., 2022). Blowing snow events are mostly common over mountains (Schmidt, 1982; Rogers and Vali, 1987; Beck et al., 2018), but
they have also been observed in the polar regions; in the Antarctic (Mann et al., 2000; Walden et al., 2003; Mahesh et al., 2003; Palm et al., 2011; Ganeshan et al., 2022) and the Arctic (Savelyev et al., 2006; Huang et al., 2008). In addition, several studies in the Arctic which observed blowing snow events relate it to sea salt aerosol emissions (Huang and Jaeglé, 2017; Frey et al., 2020; Chen et al., 2022; Gong et al., 2023), but until now no connection to INPs has been established. One way to investigate this process and its impacts is through modeling. There are already some efforts to parameterize the characteristics (e.g. size
distribution) and processes of blowing snow (e.g. sublimation, saltation) in models (Pomeroy et al., 1997; Déry and Yau, 2001; Yang and Yau, 2008; Clifton and Lehning, 2008; Chung et al., 2011; Sharma et al., 2023), in ICON, however, there is no such proposed parameterization available. Whether and to what extent blowing snow particles can have a important contribution in determining ice crystal number concentrations in Arctic mixed-phase clouds remains an open question to this day.

In the current study, we investigate the aforementioned mechanisms as potential explanations to the increased ice crystal
number concentrations over sea ice we found during the 10-year analysis of the DARDAR-Nice satellite retrieval dataset in the Arctic (Papakonstantinou-Presvelou et al., 2022). We perform sensitivity experiments with the ICON-Limited Area Model (ICON-LAM) for the time period of 21-03-2019 to 23-03-2019, which overlaps with the beginning of the Airborne measurements of radiative and turbulent FLUXes of energy and momentum in the Arctic boundary layer (AFLUX) aircraft campaign in the Arctic. We use new in-situ observations on cloud quantities measured during this campaign to consolidate our
findings, along with the satellite retrievals from the raDAR-liDAR (DARDAR) retrieval including ice number concentrations (DARDAR-Nice Sourdeval et al., 2018). The structure of the paper is the following; in Section 2.1 we describe the satellite data, in Section 2.2 we perform an evaluation of the satellite retrievals with aircraft observations, in Section 2.3 we give details on the simulation set-up and schemes used and in Section 2.4 we give a description of how the sensitivity experiments were performed. Finally, in Section 3 we report our results and in Section 4 we give a short summary and conclusions.

## 2  Data and Method

### 2.1  Satellite retrievals of ice crystal number concentration and sea ice extent

DARDAR-Nice is a satellite retrieval product of collocated lidar-radar measurements from polar orbiting satellites. It combines the CALIPSO's Cloud-Aerosol Lidar with Orthogonal Polarization (CALIOP) and the CloudSat's Cloud Profiling Radar (CPR). It applies the DARDAR variational algorithm to retrieve ice cloud properties, such as the ice water content, the ice
effective radius and the visible extinction coefficient (Delanoë and Hogan, 2008, 2010), which are then used to parameterize the particle size distribution (PSD) of ice particles (Delanoë et al., 2005). The PSD is described by a four parameter gamma



modified distribution as a function of equivalent melted diameter, from which they derive the number of particles ($N_i$) by integration from certain cutoff sizes; $5\,\mu\mathrm{m}$ and $100\,\mu\mathrm{m}$ (Sourdeval et al., 2018). The data allow the detection of both ice and liquid/mixed-phase layers, which can be later used for the elimination of uncertain measurements (e.g. ice coexisting with liquid). The DARDAR-Nice data used in this study (v.1.10) spans from mid-2006 until 2016. Thermodynamic variables, such as the temperature are taken from an auxiliary reanalysis dataset, which has been interpolated to the CPR's vertical bin (Delanoë and Hogan, 2010). The resolution of the DARDAR-Nice dataset is 1.7 km horizontally and 60 m vertically.

As another satellite product we use microwave radiometer measurements from the Advanced Microwave Scanning Radiometer on the Earth Observing System (AMSR-E) instrument onboard Aqua satellite and its successor AMSR2 onboard Shizuku. It retrieves the daily sea ice concentration at the polar regions (Arctic and Antarctic) (Spreen et al., 2008). The data are provided on a polar stereographic grid of 6.25 km resolution and are available from 2002 until present (Melsheimer and Spreen, 2019, 2020). In this case, the data are used in combination to the DARDAR-Nice dataset for the exact same period in order to distinguish clouds over sea ice and ocean in the Arctic region.

## 2.2 Evaluation of satellite results using aircraft in situ observations

The AFLUX campaign is one of the most recent campaigns that took place in the Arctic region during March-April 2019. The aim of the campaign was to investigate mixed-phase clouds and boundary-layer processes and their role to Arctic Amplification (Mech et al., 2022). The research flights took place in the vicinity of Svalbard, starting from Longyearbyen ($78°$ N, $15°$ E) and flew over the ocean, marginal ice and sea ice, between Greenland and the Fram Strait.

In this study we are interested in the microphysical cloud properties and therefore, we take advantage of this novel in-situ measurements of cloud number concentration in low-level boundary clouds. The in-situ data were collected from 3 main instruments onboard the Polar 5 aircraft; the Cloud Aerosol Spectrometer (CAS), the Cloud Imaging Probe (CIP) and the Precipitation Imaging Probe (PIP). All together they span particles with sizes from 2.8 to $6400\,\mu\mathrm{m}$. Since each instrument alone covers a certain range of particle sizes, the microphysical cloud properties (e.g. total particle number concentration, effective diameter, cloud water content) are calculated from the combined particle size distribution (Moser et al., 2023). The temporal resolution of the data is 1 Hz and the aircraft is flying with an average speed of 60 m/s, thus covering a horizontal spatial resolution of approximately 60 m. In addition to the cloud dataset, information on the temperature and the height of the flight is provided through a nose boom system from the front of the aircraft (Mech et al., 2022).

The analysis of DARDAR-Nice retrievals follows the same methodology as in Papakonstantinou-Presvelou et al. (2022). Hence, only a brief description will be given in this section and the reader is encouraged to refer to the aforementioned paper for more details. As a first step we limit the analysis to pure ice single-layer profiles, with tops below 2 km from the surface. This means that we do not account for any mixed-phase layer or any ice layer below a liquid cloud layer, because of uncertainties in those retrievals (Sourdeval et al., 2018). We additionally only choose the cloud tops, defined as the two uppermost layers of the cloud profile, to avoid falling ice crystals. We cluster the ice retrievals into three distinct temperature classes between $0°$C and $-30°$C, based on the cloud top temperature. We distinguish the profiles according to their underlying surface in two categories; sea ice and ocean. As sea ice we define a sea ice concentration of larger than 80% and as ocean what is lower than



15%. The information on the sea ice concentration is on a different grid than DARDAR-Nice , therefore an interpolation to the AMSRE grid is essential. For this study, we make use of the statistics of $N_i$ with respect to the lower cutoff size $5\,\mu\text{m}$ that corresponds to the period 2006-2016 for a modified season, February-March-April (FMA), to be closer to the time of the campaign and for the area of 60-82° N. In our previous study (Papakonstantinou-Presvelou et al., 2022) we divided the Arctic into five latitude belts of 5° each, and examined the results in each one of them, in order to avoid the bias stemming from the North-South difference in sea ice/ocean. We consider this issue here by aggregating first the median values of the distributions in each latitude belt and then for the whole Arctic.

The in-situ observations provide the temperature-binned combined distribution from the three instruments that describes all measured particles during the research flights. Therefore, the total number of particles can be obtained by integrating the size distribution starting from a certain diameter. However, several types of hydrometeors with different sizes are measured, including deliquesced aerosols, cloud droplets and ice crystals. In order to differentiate solely the ice crystals we eliminate all particles that are smaller than $50\,\mu\text{m}$, i.e. the cloud droplets and aerosols, and we assume that particles larger than this size must be ice particles, following the methodology in Moser et al. (2023). An additional limit of total cloud water content (CWC) of $10^{-5}\,\text{kg/m}^3$ is set in order to exclude precipitation, with the CWC defined as the sum of liquid and ice water content. Also, the segments of the flight that correspond to ascent, descent and turns are filtered out (for a more detailed description see Methods/Flight strategies in  Mech et al., 2022), because the quality of data there is reduced. Finally, only measurements below $2\,\text{km}$ are selected to be consistent with the previous analysis of DARDAR-Nice . The analysis of the cloud data is combined with the AMSRE sea ice concentration data, which were already collocated with the aircraft measurements and ready to use (python package ac3airborne mentioned in Mech et al., 2022). To be able to compare the DARDAR-Nice with the airborne observations, the same or similar cutoff diameter for derivation of the number of particles should be used. In this case $5\mu\text{m}$ cannot be used for the in-situ data, since this cutoff would also include cloud droplets which would bias the comparison. Therefore, we are comparing the lower cutoff $N_{i,5\mu\text{m}}$ from DARDAR-Nice which is used throughout the study to the $N_{i,50\mu\text{m}}$ from observations.

Figure 1 shows a comparison between the median number concentration of ice crystals that is observed in the satellite retrieval product DARDAR-Nice and the airborne observations collected during the AFLUX campaign. In general, DARDAR-Nice has always lower values compared to the aircraft data at all temperatures. Both datasets show median values between 1 and $10\,\text{L}^{-1}$ between 0° and −20°C. At temperatures lower than −20°C, the aircraft data show a higher value around $50\,\text{L}^{-1}$ over sea ice compared to the DARDAR-Nice value which is around $6\,\text{L}^{-1}$, while the values that correspond to the ocean are very close. The two datasets agree on the difference between sea ice and ocean, showing higher values over sea ice consistently at all temperature classes. Despite the many similarities, we should keep in mind that the two products present also some differences. First of all the difference in the cutoff size, which is higher for the aircraft observations than for DARDAR-Nice , due to the necessary omission of cloud droplets in the second case. That could lead to discrepancies between the two data. However, we also tested a higher common cutoff size of $100\,\mu\text{m}$ and the main conclusions didn't change, and the higher numbers over sea ice persisted in most of the cases but with lower concentrations overall (not shown). In addition, the aircraft data represent the measurements collected inside the cloud along the flight path, whereas DARDAR-Nice includes only single-layer cloud





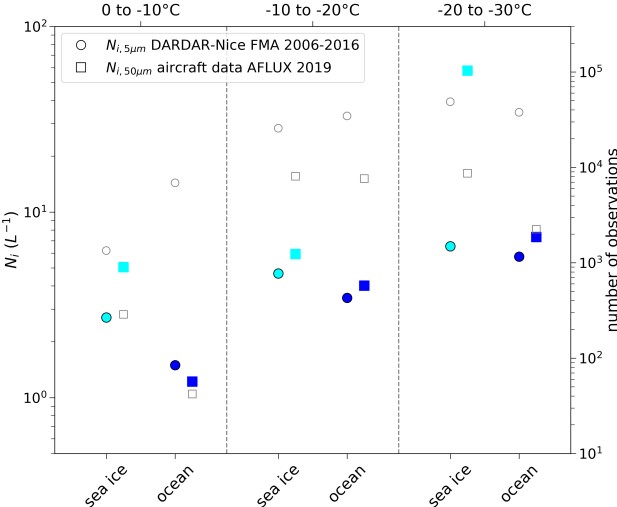

**Figure 1.** Median number concentration of ice crystals [$\mathrm{L^{-1}}$] as a function of temperature over sea ice and ocean (cyan and blue colors, respectively). Left y-axis (filled symbols) shows the satellite retrievals DARDAR-Nice as median values from the period 2006-2016 for February-March-April (FMA) and the aircraft observations during AFLUX campaign. For DARDAR-Nice the $5\,\mu m$ diameter is used as lower cutoff size, while for the aircraft data a higher cutoff of $50\,\mu m$ is used. The right y-axis (open symbols) corresponds to the number of observations used to calculate the median in each temperature bin.

top values in each AMSRE's grid cell (6.25 km). The aggregation of the aircraft data to this resolution was also performed additionally, but it lead to similar conclusions with less amount of data in each temperature bin (not shown). In conclusion, the aircraft observations corroborate what was shown in the DARDAR-Nice retrievals during 2006-2016. These satellite data are used as a reference for the model simulations in Section 3.

## 2.3 ICON-LAM simulations

The simulations in the framework of this study are conducted with the Icosahedral Non-hydrostatic model (Zängl et al., 2015) ICON version 2.6.6 (hereafter referred as ICON-v2.6.6). The ICON-LAM model is used which is basically the ICON-NWP (numerical weather prediction) in a limited area mode. We apply a two-domain set-up with one-way nesting (no feedback from the inner to the outer domain). The domain of the simulations is the Arctic region 60-90° N, 30° W-30°E (inner domain) which is enclosed in a larger domain, both of them are depicted in Figure 2. The outer domain (DOM01) has a resolution of approximately 5.0 km and the inner domain (DOM02) of 2.5 km (in ICON's terminology R2B9 and R2B10, respectively) and both have a vertical division of the atmosphere into 75 model levels. These vertical levels correspond to a terrain-following height-based coordinate. All simulations start at 20-03-2019 12:00 UTC and end at 24-03-2019 00:00 UTC. Analysis data from the Integrated Forecasting System (IFS) of the European Centre for Medium-Range Weather Forecasts (ECMWF) is used for initialization ($\approx 14$ km resolution) and for boundary conditions every 3 hours (forecast data). We consider a model spin-up time



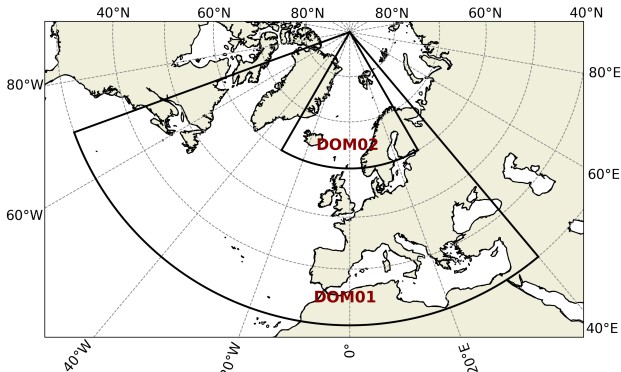

**Figure 2.** Representation of domains of the ICON simulations. DOM01 is the outer domain (R2B9, 5 km) and DOM02 the inner domain R2B10, 2.5 km).

of 12 hours and analyze model data starting from 21-03-2019 00:00 UTC (72 hours). Concerning the physics parametrizations, we make use of the two-moment cloud microphysics which calculates both the mass and number of each type of hydrometeor (Seifert and Beheng, 2006). Six types are includes; liquid, ice, snow, rain, graupel and hail. The cloud microphysics includes the heterogeneous nucleation scheme for ice described in Phillips et al. (2008), which is the default scheme in this version.

For the radiation scheme we use the ecRad radiation scheme (Hogan and Bozzo, 2018; Rieger, 2019) and for the cloud cover the "all-or-nothing" cloud cover scheme (grid scale clouds) in the inner domain while in the outer domain the diagnostic cloud cover scheme. The Tiedtke-Bechtold convection scheme (Tiedke, 1989; Bechtold et al., 2008) is completely switched off in all simulations, since we are not interested in analyzing ice crystal numbers that arise due to convection in the area.

### 2.4 Sensitivity experiments with ICON

Sensitivity experiments are used to test several hypotheses we formulated during the interpretation of the satellite retrievals DARDAR-Nice . The primary idea was to try out three main experiments and potential mechanisms that could be responsible for the difference in ice crystals between clouds over the sea ice and ocean we observed from the satellite (Papakonstantinou-Presvelou et al., 2022). These are the following:

1. potential INPs over sea ice
2. blowing snow
3. secondary ice production.

In the first experiment we want to know what is the response to the ice number if we implement an additional amount of INPs over the sea ice (e.g. due to melt ponds). In ICON-v2.6.6 the Phillips et al. (2008) parametrization for heterogeneous ice nucleation is used, which includes three main types of aerosols; dust/metallic aerosols, inorganic black carbon (soot) and 230 insoluble organics (including bacteria, leaf litters, pollen etc). These INPs act as nuclei for ice in the model through two modes;



**Table 1.** Estimated mean [minimum, maximum] organic INP concentration [L$^{-1}$] in each temperature class described in the ICON-control experiment.

| T [°C] | $N_{\mathrm{orga}}$ [L$^{-1}$] |
|---|---|
| $-5$ to $-10$ | $0.13\,[0.07, 0.18]$ |
| $-10$ to $-20$ | $0.46\,[0.18, 0.76]$ |
| $-20$ to $-30$ | $2.06\,[0.76, 3.38]$ |

i) immersion freezing starting at relatively high temperatures and ii) deposition nucleation at lower temperatures (mainly dust). The amount of aerosols that are able to act as INPs is given as three distinct initial concentrations in ICON-v2.6.6 which are then transformed to fractions of activated INPs through look-up tables. Dust and soot can initiate nucleation below $-10°$C and $-14°$C respectively, whereas organics are able to nucleate ice in relatively warm temperatures (through immersion freezing)

starting from approximately $-5°$C. In this set of simulations we focus only on one type of aerosols for simplicity; we choose the organics since they become active at warmer temperatures where the sea ice – ocean contrast in ice crystal number in the satellite retrievals was largest. The amount of organic INPs that are activated is a function of temperature. In the ICON-control simulation the mean, minimum and maximum organic INP concentration has been estimated for the three temperature classes we are looking at (see Table 1). In each sensitivity experiment, we keep the organic INPs constant everywhere and over sea ice

we introduce an additional amount, which is enhanced by a factor of 5, 50 and 100 each time and investigate the effect of this change. The definition of sea ice used in all experiments is the same as described in Section 2.2.

     The second experiment investigates the blowing snow effect, i.e. if blown ice particles from the frozen surface of sea ice are capable of enhancing the ice crystal concentration. There are two pathways through which this can happen. Either the blowing snow breaks down to several ice particles that are then transported to the cloud level and contribute to the ice number

itself, or there is an INP source within the blowing snow which can act as ice nuclei when transported to the cloud level. Since there is no parametrization in ICON-v2.6.6 that describes any of these processes, in this study we implemented a rather simple approach of direct transport of ice crystals without any interference of aerosols. This is to test whether an enhancement of cloud ice crystals due to blowing snow may explain the satellite observations, no matter the exact mechanism that lies behind this process. Blowing snow events are triggered when strong winds are present. In order to parameterize this process in ICON we

define a wind speed threshold that allows the particles to be blown into the atmosphere. Several studies propose a wind speed threshold which is usually higher than 7 m/s (Walden et al., 2003; Mahesh et al., 2003; Huang and Jaeglé, 2017; Ganeshan et al., 2022). Other studies (Chung et al., 2011; Frey et al., 2020) estimate a wind threshold using a temperature-dependent relationship proposed by Li and Pomeroy (1997):

$$U_t = 6.98\,\mathrm{m\,s}^{-1} + 0.0033\,\mathrm{m\,s}^{-1}\,\mathrm{K}^{-2}\,(T_\mathrm{a} - 245.9\,K)^2 \tag{1}$$



where $T_a$ is the 2 m temperature in K. Here, we use this relationship to define the wind threshold and introduce an amount of
blown ice particles over the sea ice. Following the methodology from Georgakaki et al. (2022) we use a constant source of ice
crystals from blowing snow, from $1\,\mathrm{L}^{-1}$ to $3\ \mathrm{L}^{-1}$ (blowing snow rates ranging from $0.05\,\mathrm{L}^{-1}\mathrm{s}^{-1}$ to $0.15\,\mathrm{L}^{-1}\mathrm{s}^{-1}$), which we
implement in the model's first level (surface) all over the domain where there is sea ice. We assume a spherical size for the
particles with a mean diameter of $100\,\mu\mathrm{m}$ (Schmidt, 1982; Geerts et al., 2015; Georgakaki et al., 2022; Sharma et al., 2023).

In the third experiment we test whether secondary ice production (SIP) could provide a valid explanation on the high ice
numbers and/or the difference between the surfaces found before. There are plenty of mechanisms of SIP in nature, nevertheless
in ICON-v2.6.6 only one main mechanism is implemented and that's the Hallet-Mossop process (Hallett and Mossop, 1974).
This process acts only at relatively warm temperatures, between $-3^\circ$C and $-8^\circ$C. In this temperature regime the model
calculates a number of multiplied particles, which is then added to the ice number and mass. The efficiency of this process is

given through a predefined coefficient, which in this version is set to $3.5\cdot10^8$. In this set of experiments we investigate the
effect of the SIP process on the ice numbers over sea ice and ocean, by changing this coefficient. The experiments shown here
concern multiplying factors of $1/2$, 2 and 10, i.e. the assumption that the process is half, double or ten times more efficient as
in the default setting.

    The analysis of the model simulations closely mirrors the approach used for the satellite retrievals. A comprehensive de-

scription of the methodology used to compare model data with satellite observations—such as the calculation of ice crystal
number concentrations relative to a specified cutoff diameter—can be found in Appendix A.

## 3   Results

The results that follow refer to the three sensitivity experiments that were performed to investigate the potential cause/s behind
the difference in ice crystal numbers between sea ice and ocean as found from the satellite observations. The first experiment

testing the organic INP sensitivity to ice crystal numbers is depicted in Figure 3. The different factors show how much more
organic INPs over sea ice are added compared to over the ocean. DARDAR-Nice appears to overestimate the model's control
ice crystal number concentration through the temperature range (Fig. 3, left panel). In Sourdeval et al. (2018) they also found
an overestimation of $N_{\mathrm{i},5\mu\mathrm{m}}$ due to the misinterpretation of the shape of the PSD at warm temperatures compared to in-situ
measurements. On the other hand, active remote sensing provides information from all parts of the atmospheric column and is

not biased by the underlying surface. Thus, we cannot be sure about the absolute value of $N_\mathrm{i}$, but as a more reliable reference
we consider the difference that appears in $N_\mathrm{i}$ values over sea ice and ocean. To evaluate the contrast between the two surfaces
we calculated the fraction of the $N_\mathrm{i}$ over sea ice divided by the $N_\mathrm{i}$ over ocean (Fig. 3, right panel).

    In Fig. 3, as expected, the more organics are added, the more ice crystals are nucleated due to heterogeneous nucleation.
In addition, the ice nucleation is temperature dependent, and more effective as the temperature drops. In particular, when

adding 5 times more organics than the control concentration over sea ice, the concentration of ice crystals increases a little in
all temperature bins, but is way lower than in DARDAR-Nice . However, the ratio of ice crystals over sea ice vs. over open
ocean is in agreement to the fraction by DARDAR-Nice below $-10^\circ$C (Fig. 3, right panel). In the case of further increasing



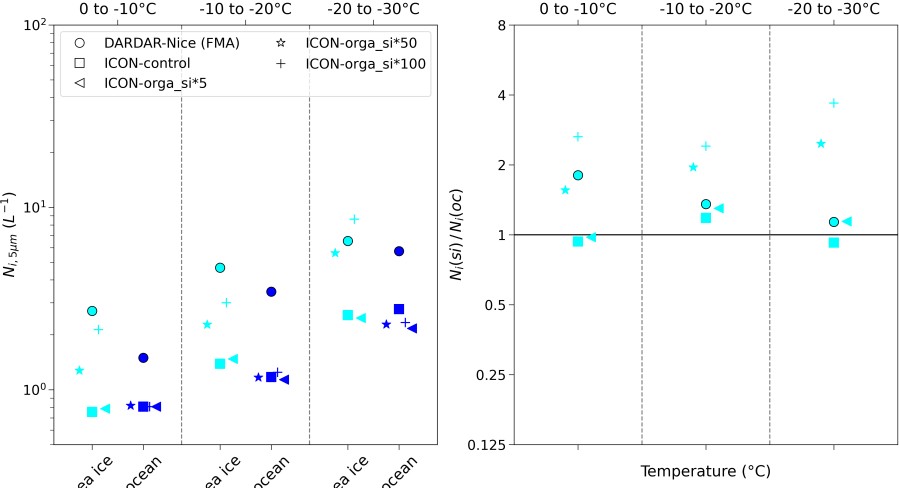

**Figure 3.** Left: median number concentration of ice crystals [L$^{-1}$] that are larger than $5\mu$m as a function of temperature over sea ice and ocean (cyan and blue colors, respectively). Right: ratio of $N_i$ over sea ice vs. over open ocean. The horizontal line means that there is no difference between the two. The different symbols represent: the satellite retrievals DARDAR-Nice from the period 2006-2016 for February-March-April (FMA) and the sensitivity experiments with ICON model (control, orga_six5, orga_six50 and orga_six100), where the numbers represent how much more organics are added over sea ice compared to over ocean.

concentrations of organic INPs over sea ice by a factor of 50 and 100, the concentrations become closer to DARDAR-Nice and in some cases overestimate them. However, for such assumptions, the ratio in $N_i$ between the two surfaces is overestimated

by up to 4 times in the lowest temperature bin. Therefore, increasing the organic INPs by a factor of 5 could explain the contrast in $N_i$ between sea ice and ocean below $-10°$C, but not above this temperature. There, an even higher concentration of INPs could explain the difference, but such an assumption at the same time would not allow to explain the ratio for the other temperature ranges. Sensitivity experiments with higher concentrations of organic INPs were also tested, but resulted to even higher concentrations of ice and higher $N_i$ fractions between the surfaces and thus are not shown in the context of this study.

Blowing snow particles in the form of ice can also enhance the ice crystal numbers given certain wind conditions as shown in Figure 4. Several numbers of blown particles were tested from 1-100 L$^{-1}$ (ranging from 0.05 L$^{-1}$s$^{-1}$ to 5 L$^{-1}$s$^{-1}$) but the three most relevant experiments are shown here. Adding 1 L$^{-1}$ and 2 L$^{-1}$ leads to a rather small increase in $N_i$, but adding 3 L$^{-1}$ leads to overestimation of $N_i$ compared to DARDAR-Nice at all temperatures. As for the ratio in $N_i$ between sea ice and ocean, the first experiment underestimates this ratio above $-10°$C, agrees approximately with the satellite result between $-10$

and $-20°$C and shows a little overestimation below $-20°$C. The second experiment shows a similar behavior with slightly higher values in all temperatures, while the third always overestimates by far the difference. In conclusion, an amount of blowing snow of about 1 L$^{-1}$ could explain the difference between sea ice and ocean in temperatures below $-10°$C, but not above this.



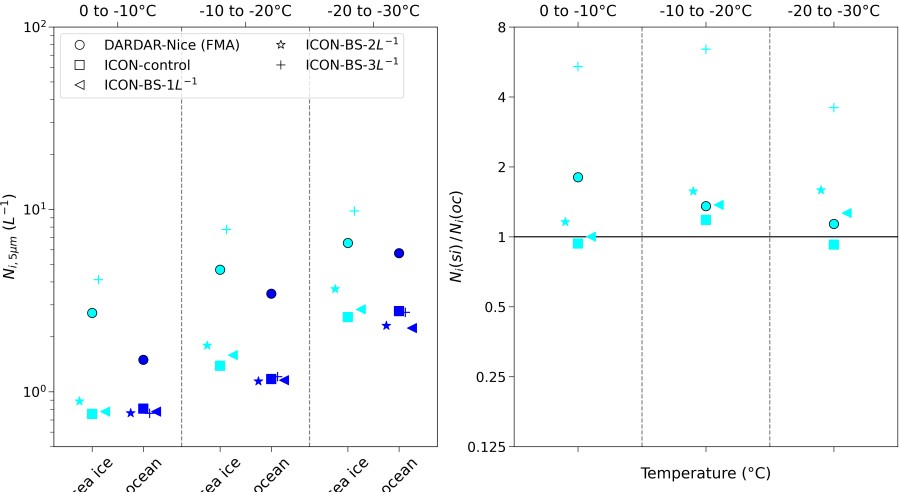

**Figure 4.** Same as Figure 3, but here the different symbols represent: the satellite retrievals DARDAR-Nice from the period 2006-2016 for February-March-April (FMA) and the sensitivity experiments with ICON model (control, BS-1L$^{-1}$, BS-2L$^{-1}$ and BS-3L$^{-1}$) where the numbers represent how much blowing snow particles is added over sea ice.

Secondary ice production experiments are shown in Figure 5. The effect of the change of this process can be detected only in
the first temperature class, between 0 and $-10°$C. The enhancement of SIP in ICON affects more the $N_\mathrm{i}$ over ocean than the $N_\mathrm{i}$
over sea ice and leads to higher numbers over ocean. This can be seen in both the experiments where SIP process was enhanced
(SIPx2 and SIPx10). That suggests an opposite result to what DARDAR-Nice shows. We also tested the case of decreased SIP
in ICON (SIPx1/2), where the results show that the $N_\mathrm{i}$ over ocean decreases. Further sensitivity experiments with decreased
SIP did not improve the results (not shown). In conclusion, enhanced SIP in ICON cannot explain the difference between sea
ice and ocean we found from the satellite retrievals DARDAR-Nice .

## 4   Summary and Conclusions

In this study we were interested in ice cloud microphysics in the Arctic, specifically their differences between sea ice and open
ocean. The motivation is that as sea ice continuously retreats in a warming climate, systematic differences may imply a relevant
climate feedback. The study follows a previous analysis of satellite retrievals of the ice crystal number concentration in Arctic
boundary-layer ice clouds (Papakonstantinou-Presvelou et al., 2022). That study found that there are systematically more ice
crystals over sea ice than over open ocean in the Arctic, for temperatures between $0°$ and $-30°$C. In the present study, we seek
causal explanations for these differences using a kilometer-resolution atmospheric model.

We perform sensitivity experiments with the ICON model in order to test possible explanations for the enhanced ice crystal
numbers, using the satellite results from DARDAR-Nice as a reference. All simulations are performed over the Arctic in a pe-
riod of 3 days, that overlaps with the AFLUX observational campaign. We compare our satellite retrievals to new observations




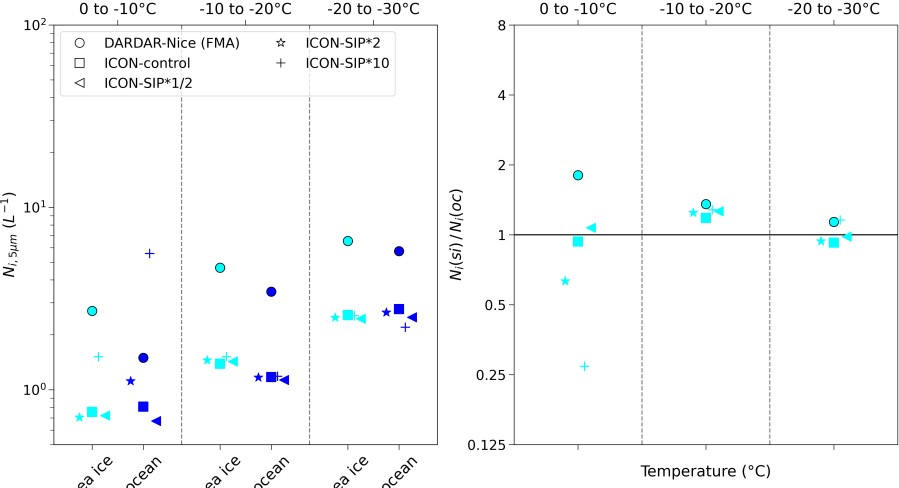

**Figure 5.** Same as Figure 3, but here the different symbols represent: the satellite retrievals DARDAR-Nice from the period 2006-2016 for February-March-April (FMA) and the sensitivity experiments with ICON model (control, SIPx1/2, SIPx2, SIPx10) where the numbers represent how much more/less efficient is the secondary ice production.

collected during that campaign and we find a general agreement when it comes to the positive difference of ice number concentration over sea ice compared to the ocean. The three possible mechanisms we are examining in the sensitivity experiments are an INP source over sea ice, blowing snow and secondary ice production. For the first experiment we change the concentrations of organic INPs as imposed as boundary condition to the ICON simulations. For the last, we scale the efficiency of SIP that is

parameterized as a Hallett-Mossop process. For the blowing snow experiment we implement a new routine in ICON that adds number and mass of ice in the first model layer depending on the wind conditions. Our results show that increased organic INPs over sea ice and blowing snow are both possible explanations, according to the model, for the positive difference over sea ice. In contrast, SIP is not able to reproduce this difference.

      In particular, the experiment with organic INPs enhanced by a factor of 5 over sea ice (orga_six5) agrees with DARDAR-

Nice below $-10°$C. Above this temperature a much higher INP concentration would be necessary (a factor between 50 and 100), which would then lead to a disagreement at lower temperatures. According to Table 1, the orga_six50 would imply an INP concentration of more than $3.5\,\mathrm{L}^{-1}$ at $-5°$C, which is higher than what is shown by measurements of INPs in various locations in Petters and Wright (2015). Between $-10°$C and $-20°$C, the orga_six5 experiment gives more realistic concentrations of INPs starting from $0.9\,\mathrm{L}^{-1}$ at $-10°$C up to $3.8\,\mathrm{L}^{-1}$ at $-20°$C, closer to the spectrum defined in Petters and Wright (2015) and

below $-20°$C the numbers reach up to $15\,\mathrm{L}^{-1}$ which is even lower than that. However, the INP concentrations required for this explanation are substantially larger than what has been reported from measurement campaigns in the Arctic. According to these studies INP numbers are up to the order of $10^{-2}\,\mathrm{L}^{-1}$ for warmer temperatures, above $-15°$C (Wex et al., 2019) and reach a little higher than $0.1\,\mathrm{L}^{-1}$ for lower temperatures, down to $-25°$C (Hartmann et al., 2020). This discrepancy could possibly stem from the fact that the Phillips et al. (2008) INP parameterization in ICON is based on observations from multiple



campaigns and is not specifically designed for Arctic INP distributions. It might also be that the in situ observations cited above are not representative for INPs over sea ice in the Arctic.

Concerning the blowing snow experiment, our results show that blown ice particles are able to reproduce the difference in ice number over sea ice below $-10°$C. A quantity of $1\,\mathrm{L^{-1}}$ ($0.05\,\mathrm{L^{-1}s^{-1}}$ rate of emission of blowing snow) would be sufficient to explain the difference at these lower temperatures. Above $-10°$C a higher quantity would be necessary (between $2\,\mathrm{L^{-1}}$ and

$3\,\mathrm{L^{-1}}$), but this would lead to overestimation of the difference at lower temperatures. In Georgakaki et al. (2022) they suggested a much larger quantity of blowing snow particles ($100\,\mathrm{L^{-1}}$) than what we find, but in their case they examined alpine mixed-phase clouds and their experiments were performed with a different model and conditions. However, our implementation of blowing snow emission in ICON is a rather simple approach and therefore it cannot explain all aspects of the phenomenon, highlighting the need for a complete parametrization in ICON.

Overall, these findings suggest that INPs and wind-blown particles from the frozen sea ice surface can be determinants for the ice number contrast between clouds over sea ice and ocean, while SIP according to our results cannot. Understanding the role of these particles is critical for improving cloud microphysics parameterizations in climate models, enhancing in turn the accuracy of Arctic climate predictions and our understanding of Arctic amplification. Since sea ice will continue to retreat in a warming climate, ice clouds may consist of fewer particles. It can be expected that this implies less reflective clouds, and thus

a positive feedback.

*Code and data availability.* The DARDAR-Nice data is provided by the AERIS/ICARE Data and Services Center upon request. The AMSRE and AMSR2 ASI sea ice concentration data is publicly available through PANGAEA Data Publisher for Earth & Environmental Science (Melsheimer and Spreen, 2019, 2020). The DLR in-situ cloud measurements during the AFLUX Arctic airborne campaign are publicly available in PANGAEA (Moser and Voigt, 2022). The data can be accessed and processed using the python package ac3airborne

https://igmk.github.io/how_to_ac3airborne/intro.html. The ICON model is freely available at http://www.icon-model.org/ (ICON partnership (DWD and MPI-M and DKRZ and KIT and C2SM), 2024).

## Appendix A: ICON simulations: Analysis & calculation of number of particles from a cutoff diameter

In the analysis of the ICON data we stick to the same methodology as for the satellite retrievals. However, since the differences of the two products are high, this is not always possible and the closest approach is used. We distinguish only single-layer

clouds and its cloud tops. To do so, we define as separate clouds those that are separated by at least one vertical model level. ICON simulates all kinds of clouds, so in order to be as close as possible to the ice-type followed in DARDAR-Nice , we define as cloud what has an ice mass ($q_i$) of at least $10^{-5}\,\mathrm{kg^{-1}}$. We make sure we don't analyze liquid pixels (that exist below or above) by creating a liquid cloud mask, using the same threshold for liquid mass ($q_c$). We further include an additional threshold of ice fraction, which is the fraction of ice to the total mass of the cloud pixel (liquid+ice) to be higher than 0.9. Eventually, we need

to set the same cutoff diameter to be able to compare correctly the numbers from the model and the satellite data (available



cutoff diameters: 5 $\mu$m and 100 $\mu$m). Thus, we calculate the number of ice crystals with respect to those cutoff sizes using the particle size distribution of particles in ICON.

The two-moment microphysical parameterization in ICON (Seifert and Beheng, 2006) uses a generalized $\Gamma$-distribution to describe the PSD of each type of hydrometeors (equation 79 in Seifert and Beheng, 2006):

$$f(x) = Ax^{\nu} exp(-\lambda x^{\mu}) \tag{A1}$$

where x is the particle mass and A, $\lambda$ are coefficients dependent on the known quantities, such as the number and mass densities and and $\nu$ and $\mu$ are constant numbers. We can write the PSD as a function of diameter instead of mass, using the power law (equation 32 in Seifert and Beheng, 2006):

$$D = ax^{b} \tag{A2}$$

where a, b constant coefficients. Thus, after several calculations the PSD as a function of D is:

$$f(D) = A_D D^{\nu_D} exp(-\lambda_D D^{\mu_D}) \tag{A3}$$

Following the methodology to transform the PSD as a function of one descriptor to another in Petty and Huang (2011) (eq. 52-54) and similarly to what was described in Kretzschmar et al. (2020) (Appendix B, eq. B4-B7), but there for particle radius, we can write accordingly:

$$A_D D^{\nu_D} exp(-\lambda_D D^{\mu_D}) = A[x(D)]^{\nu} exp(-\lambda[x(D)]^{\mu}) \frac{dx}{dD} \tag{A4}$$

From differentiation of A2 we get:

$$\frac{dx}{dD} = \frac{1}{b}\left(\frac{1}{a}\right)^{1/b} D^{1/b-1} \tag{A5}$$

The substitution of A2 and A5 into A4 leads to:

$$A_D D^{\nu_D} exp(-\lambda_D D^{\mu_D}) = A\left(\frac{D}{a}\right)^{\nu/b} exp\left[-\lambda\left(\frac{D}{a}\right)^{\mu/b}\right] \frac{1}{b}\left(\frac{1}{a}\right)^{1/b} D^{1/b-1} \tag{A6}$$

with the converted PSD parameters being:

$$
\begin{aligned}
A_D &= \frac{A}{b}\left(\frac{1}{a}\right)^{\frac{\nu+1}{b}} \\
\nu_D &= \frac{\nu+1-b}{b} \\
\lambda_D &= \lambda\left(\frac{1}{a}\right)^{\frac{\mu}{b}} \\
\mu_D &= \frac{\mu}{b}
\end{aligned}
\tag{A7}
$$

The distribution as a function of diameter after calculations is simplified to this formula:

$$f(D) = B \cdot D^n \cdot exp\left\{-C \cdot D^m\right\} \tag{A8}$$



**Table A1.** Hydrometeor parameters defined in the two-moment microphysics scheme in ICON-v2.6.6 .

|  | a (m/kg$^b$) | b | $\nu$ | $\mu$ | $\bar{x}_{min}$(kg) | $\bar{x}_{max}$(kg) |
|---|---|---|---|---|---|---|
| cloud droplets | 0.124 | 1/3 | 1.0 | 1.0 | $4.2 \cdot 10^{-15}$ | $2.6 \cdot 10^{-10}$ |
| ice crystals | 0.835 | 0.39 | 0 | 1/3 | $1.0 \cdot 10^{-12}$ | $1.0 \cdot 10^{-5}$ |
| raindrops | 0.124 | 1/3 | 0 | 1/3 | $2.6 \cdot 10^{-10}$ | $3.0 \cdot 10^{-6}$ |
| snowflakes | 5.13 | 0.5 | 0 | 0.5 | $1 \cdot 10^{-10}$ | $2 \cdot 10^{-5}$ |
| graupel | 0.142 | 0.314 | 1.0 | 1/3 | $4.19 \cdot 10^{-9}$ | $5.3 \cdot 10^{-4}$ |
| hail | 0.137 | 1/3 | 1.0 | 1/3 | $2.6 \cdot 10^{-9}$ | $5 \cdot 10^{-3}$ |

where:

$$B = \frac{\mu}{a^{\frac{\nu+1}{b}}b} \cdot \frac{\Gamma\left(\frac{\nu+2}{\mu}\right)^{\nu+1}}{\Gamma\left(\frac{\nu+1}{\mu}\right)^{\nu+2}} \cdot \bar{x}^{-(\nu+1)} \cdot N$$

$$C = \frac{1}{a^{\frac{\mu}{b}}} \cdot \left[\frac{\Gamma\left(\frac{\nu+2}{\mu}\right)}{\Gamma\left(\frac{\nu+1}{\mu}\right)}\right]^{\mu} \cdot \bar{x}^{-\mu} \tag{A9}$$

$$n = \frac{\nu+1-b}{b}$$

$$m = \frac{\mu}{b}$$

where $\bar{x}$ is the mean mass of the particles, which is basically the fraction of mass (L) divided by number (N). In order to obtain the number of particles with respect to a cutoff diameter, we need to integrate the PSD from this size. The final mathematical expression is:

$$N_{D_c} = \int\limits_{D_c}^{+\infty} f(D)dD = N \cdot \frac{\Gamma\left(\frac{\nu+1}{\mu}, \frac{1}{a^{\frac{\mu}{b}}} \cdot \left[\frac{\Gamma\left(\frac{\nu+2}{\mu}\right)}{\Gamma\left(\frac{\nu+1}{\mu}\right)}\right]^{\mu} \cdot \bar{x}^{-\mu} D_c^{\frac{\mu}{b}}\right)}{\Gamma\left(\frac{\nu+1}{\mu}\right)} \tag{A10}$$

with B, C, $\mu$, $\nu$ numbers that are dependent on known model quantities or constants. Eventually using A10 we can calculate the number of particles that are larger of a certain cut-off diameter for any type of hydrometeor. The parameters for each hydrometeor that are used in the current model version (v. 2.6.6) are shown in the Table A1.

*Author contributions.* IP and JQ conceived the study. IP ran the model simulations, performed the analysis and wrote the article. JQ gave his expertise throughout the study and contributed to the improvement of the article.



*Competing interests.* Johannes Quaas is a member of the editorial board of ACP.

*Acknowledgements.* The authors gratefully acknowledge the funding by the Deutsche Forschungsgemeinschaft (DFG, German Research Foundation) for the projects: i) Projektnummer 268020496–TRR 172, within the Transregional Collaborative Research Center "ArctiC Amplification: Climate Relevant Atmospheric and SurfaCe Processes, and Feedback Mechanisms $(AC)^3$" and ii) CloudTrend (GZ QU 311/28-1). This work used resources of the Deutsches Klimarechenzentrum (DKRZ) granted by its Scientific Steering Committee (WLA) under

project ID 1143. This study would not have been possible without the contributions of our colleagues Jan Kretzschmar, Sajede Marjani, and Sabine Hörnig, whose expertise in ICON modeling was instrumental in the implementation of this work. We also extend our gratitude to Axel Seifert for the valuable discussions on the two-moment scheme in ICON. We are particularly grateful to Paraskevi Georgakaki and Tereza Kiszler for their insightful discussions regarding our results. Lastly, we would like to thank Manuel Moser for providing the observations from the AFLUX campaign and for his guidance on the proper use of the data.





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
