# Peer review of "Sensitivity experiments with ICON-LAM to test probable explanations for higher ice crystal number over Arctic sea ice vs. ocean"

_EGUsphere, 2024_

## Referee Comment (RC1)

**Review of Papakonstantinou-Presvelou and Quaas:** Sensitivity experiments with ICON-LAM to test probable explanations for higher ice crystal number over Arctic sea ice vs. ocean

**Summary and Major Comments:**

This study is a follow-up to a previous study by the same authors which found that ice crystal numbers, as estimated by DARDAR-Nice, are higher on average in Arctic low clouds occurring over sea ice than those occurring over open ocean. They test three hypotheses for why this might be: that there are more INPs over sea ice than over open ocean, that blowing snow contributes cloud ice particles to low clouds over sea ice, and that secondary ice production operates differently over the open ocean and over sea ice. They conclude that, because secondary ice production increases ice crystal numbers more over open ocean than over sea ice, blowing snow and more INPs over sea ice are the only tested processes that can explain the discrepancy in ice crystal numbers between clouds over open ocean and over sea ice.

Unfortunately, I am completely unconvinced of the authors' conclusions. They test the role of secondary ice production by including one secondary ice production process (Hallett Mossop rime-splintering) that operates within a narrower range of temperatures than the range that many observational studies have found secondary ice processes to operate on. They do not provide an explanation for why they expected that Hallett Mossop rime-splintering might behave differently in clouds over open ocean and over sea ice (and why it ultimately does but that the difference is in the wrong direction to explain the discrepancy).

They simulate a source of INPs over sea ice by multiplying the organic INP concentrations in ICON by constant (and large) factors over sea ice. They choose to do this with the organic INPs because they activate at high temperatures and the discrepancy in ice crystal numbers between low clouds over sea ice and over open ocean is largest at high temperatures (lines 235-237). Thus, they choose the INP properties that would best resolve the discrepancy, not ones that they expect based on observations or physical reasoning. In their literature review, they mention that there have been measurements of biogenic aerosol that can serve as INPs over both open ocean and melting ice, but they did not provide any evidence for there being larger concentrations of those particles over melting sea ice than over open ocean. Furthermore, they increased the INP concentrations uniformly over all sea ice, without taking into account where there was actually melting.

The approach for testing blowing snow is also simplified because the authors assume snow particles directly become cloud ice particles which has not been suggested by the studies on this topic (as the authors explain in lines 242-247). However, this test does have more solid physical reasoning behind it than the increase in organic INPs over sea ice.

For all three tests, the authors use several scaling factors for each parameterization. Taking into account the simplicity of the parameterizations used combined with the use of arbitrary scaling factors, it is very difficult to believe that the physical processes the authors intend to test are being well-represented.

I also don't think that the explanations the authors investigated are the only ones capable of explaining the discrepancy they find in DARDAR-Nice. They have not considered that there may be differences in the dynamical environments between the clouds over open ocean and those over sea ice. The authors acknowledge that clouds over sea ice and over open ocean have different morphologies (lines 49-50), wherein clouds over open ocean are more likely to be stratocumulus and clouds over sea ice are more likely to be thin stratus. Stratocumulus clouds have more turbulence in them which can alter liquid and ice formation processes, and the interactions between liquid and ice particles. However, that turbulence is not resolved in the kilometer-scale model that the authors use and thus their analysis is not sensitive to this alternative explanation.

Looking to the dynamics makes sense because even if INP sources are different at the surface over open ocean and over sea ice, the particles can be transported over long distances and the INPs in the clouds may not perfectly reflect those at the surface. Thus, clouds over open ocean can have sea ice INPs in them, and clouds over sea ice can have open ocean INPs in them. The dynamics are largely controlled by the fluxes at the sea surface which change instantaneously as you go between sea ice and open ocean.

Given the simplicity in the microphysics parameterizations used and the limitations of kilometer-scale models in representing the dynamics influencing microphysics in low clouds, I strongly recommend that the authors rely more strongly on the observations available to sort through their hypotheses for why ice crystal numbers are larger in the low clouds over sea ice.

The authors only use the aircraft observations to compute ice crystal numbers, and the observations fall

out of their analysis after Figure 1 although they would also be relevant for Figures 3-5. It would be better if they looked at the observed ice crystal size distributions. Then they could see the differences both in ice crystal number and ice crystal size between clouds over sea ice and clouds over open ocean, and that could give some clues as to what the ice formation mechanisms are in each population of clouds. Furthermore, they can use the observed size distributions to evaluate the assumed size distributions in DARDAR-Nice, which may not be equally appropriate for the clouds over open ocean and over sea ice. Additionally, the authors should give an explanation for the discrepancy between ice crystal numbers from aircraft and from DARDAR-Nice over sea ice at temperatures between -20 and -30 C (Figure 1).

The authors can use satellite estimates of melt pond fraction in conjunction with observations from both aircraft and DARDAR-Nice to see if ice crystal numbers are larger over melt ponds, and to test that idea that melt ponds are a source of INPs over sea ice. If blowing snow is an important contributor to ice crystal numbers over sea ice, then the ice crystal numbers from both aircraft and DARDAR-Nice should decrease with height. Additionally, the authors can look for relationships between surface wind speeds from reanalysis and observed ice crystal numbers to further investigate blowing snow.

The observations that ice crystal numbers are larger in low Arctic clouds over sea ice than over open ocean, contrary to expectations, is interesting, and the authors can make a nice study in sorting out the possible explanations. However, I do not believe that they have considered enough hypotheses here, or done a thorough enough investigation of the processes that they have considered. I look forward to a future version of this study that uses the full suite of observations and simulations available to build a convincing and consistent physical picture of how ice particles form and/or evolve differently in the two populations of clouds studied (low Arctic clouds over sea ice and over open ocean).

---

## Referee Comment (RC2)

The presented study follows up the research by the same authors with observations showing counterintuitively more ice crystals over sea ice than over open ocean in Arctic ice boundary-layer clouds. The presented study aims to explore the potential causes behind this contrast by comparing the satellite observation results from DARDAR-Nice and ICON-LAM atmospheric model. Three hypotheses were tested to explain the relative difference in ice crystal numbers: 1) potentially more INPs over sea ice than over open ocean; 2) difference in contribution from blowing snow; and 3) difference in secondary ice production (SIP). The authors conclude that INPs and blowing snow could determine the difference in observed ice crystal numbers over sea ice and open ocean, while SIP is not a determinant in this contrast.

Despite the intriguing scientific question of what drives this unexpected observation, the manuscript faces major shortcomings in several critical areas, which make its conclusions scientifically unconvincing:

1. Inadequate or over-simplified model representations

    Insufficient and not well-presented results to support the interpretations and conclusions due to inappropriate selection of tool, i.e., the km-scale ICON model, which does not explain the atmospheric dynamics that are likely responsible (and definitely worth considered as one hypothesis or impacting factor) for the observed difference in ice crystal numbers over different landscape based on the authors' original hypotheses. E.g., turbulence that forms ice differently (as stressed by referee 1), atmospheric transport of aerosols that could shift the INP abundance in ice boundary-layer clouds, changing in leading mechanisms of SIP due to the differences in vertical temperature and saturation profile, etc. In addition, within the chosen method, the setup of models (e.g., scaling parameters) are usually selected arbitrarily without proper reasoning or citations and were largely simplified, leading to subjective and less representative results for the studied scenario. I strongly support the idea from referee 1 to dig into the observational data from different dimensions instead, not limited to the field observations but maybe also reanalysis data, including (in addition to particle size distribution suggested by referee 1) surface wind field for further investigation of blowing snow and airmass history analysis by backward trajectories to find out the sources and sinks of aerosols (INPs) during the investigated period.

2. Lack of detailed scientific explanations

    The Result section focused on describing the obvious statistics from the presented figures without further digging into the science or physical mechanisms behind the "phenomenon". In general, lacking solid scientific explanation makes the

presented paper more like an experimental report rather than an academic paper. Many open questions remain unexplained, including but not limiting to e.g., what could be the extra INP sources over sea ice? Does organic INPs representative enough in the Arctic environment? What are the driving forces behind more blowing snow/SIP over ice compared to the ocean? Why the contrast in ice crystal numbers different in different setup and temperature ranges? Etc. The storyline is completely unclear without addressing these questions arise from your results.

3. Selective reporting of results

Even for the presented results, they are sometimes not accurately demonstrated, or the authors seem to manually pick the results that are more supportive or significant to explain their hypotheses. It was mentioned several times some results are not shown when they are not significant or against the hypotheses (e.g., lines 194, 294, and 309). In my opinion, it is unfair to conclude directly without showing the results because the readers should reserve the rights to make judgement out of an unbiased dataset provided by authors.

4. Insufficient and misapplied citations

As stressed above, in the Method and Result sections regarding ICON model setup, many configurations/parameters were announced without convincing reasonings or appropriate citations of relevant work. In addition, the author did not credit the proper citation of Arctic data, e.g., in lines 333-334, Petters and Wright (2015) was wrongly cited for comparing the Arctic INP data, which represents the global precipitation samples.

5. Missing Discussion Section

A Discussion section is missing, which should necessarily illustrate the limitation of the presented study, including e.g., the experimental setup (e.g., selection of scaling parameters, shortage of data due to very wide temperature bins); factors/hypotheses that are not examined but could have influence in the resulting difference in observed ice crystal numbers.

Regarding figures and data presentation, Figures 3–5 could be improved substantially:

- Consider using more effective color scales and symbols to make the data clearer.

- Eliminate redundant information or merge panels if they convey similar points; for instance, the right panels might be replaced with a more concise statistical significance test.

- Narrowing the temperature bins or rearranging the experiments into a single figure could offer a clearer comparative perspective.

Finally, the manuscript's language and structure would benefit from a thorough review, ideally by a native English speaker. Several informal expressions should be replaced with more precise academic language, unnecessary repetitions can be removed, and grammar/spelling errors need to be addressed. The paper would also benefit from overall conciseness, especially in the Introduction.

I look forward to following up this interesting research topic and the improved version of this paper with more convincing scientific storyline and evidence, clearer and more concise demonstration of results, and more professional and precise language.

---

## Author Comment (AC2)

**Reviewer #2**

The presented study follows up the research by the same authors with observations showing counterintuitively more ice crystals over sea ice than over open ocean in Arctic ice boundary-layer clouds. The presented study aims to explore the potential causes behind this contrast by comparing the satellite observation results from DARDAR-Nice and ICON-LAM atmospheric model. Three hypotheses were tested to explain the relative difference in ice crystal numbers: 1) potentially more INPs over sea ice than over open ocean; 2) difference in contribution from blowing snow; and 3) difference in secondary ice production (SIP). The authors conclude that INPs and blowing snow could determine the difference in observed ice crystal numbers over sea ice and open ocean, while SIP is not a determinant in this contrast.

*We thank the reviewer for kindly assessing our manuscript and for this very good summary.*

Despite the intriguing scientific question of what drives this unexpected observation, the manuscript faces major shortcomings in several critical areas, which make its conclusions scientifically unconvincing:

1. Inadequate or over-simplified model representations

Insufficient and not well-presented results to support the interpretations and conclusions due to inappropriate selection of tool, i.e., the km-scale ICON model, which does not explain the atmospheric dynamics that are likely responsible (and definitely worth considered as one hypothesis or impacting factor) for the observed difference in ice crystal numbers over different landscape based on the authors' original hypotheses. E.g., turbulence that forms ice differently (as stressed by referee 1), atmospheric transport of aerosols that could shift the INP abundance in ice boundary-layer clouds, changing in leading mechanisms of SIP due to the differences in vertical temperature and saturation profile, etc. In addition, within the chosen method, the setup of models (e.g., scaling parameters) are usually selected arbitrarily without proper reasoning or citations and were largely simplified, leading to subjective and less representative results for the studied scenario. I strongly support the idea from referee 1 to dig into the observational data from different dimensions instead, not limited to the field observations but maybe also reanalysis data, including (in addition to particle size distribution suggested by referee 1) surface wind field for further investigation of blowing snow and airmass history analysis by backward trajectories to find out the sources and sinks of aerosols (INPs) during the investigated period.

*We thank the reviewer for elaborating on comments raised by Reviewer #1. The first suggestion is that dynamics may lead to differences in ice formation over sea ice vs. ocean. This is the analysis in our control simulation. We clarify this point in the revision of the manuscript. The second point is that the scaling is considered arbitrary. We think, however, that this entirely makes sense; if sensitivity studies with even a very large perturbation to a specific process do not allow to attribute the detected difference in Ni over sea ice vs. ocean, then a smaller – potentially more realistic – perturbation would certainly not allow for such a conclusion. We clarified this point in the revision. The third aspect is the suggestion to analyze in situ data. However, it is the fact that there are only sparse observations in the Arctic, and that conclusions are partly contradictory, which call for the use of remote sensing for observations-based analysis (which is the basis for the present study). In either case, modelling – as done in this study – is required for attribution of causes to detection of effects. We clarify this now at the beginning of our revised manuscript. The reviewer further suggests to analyze reanalysis data for potential INP sources as well as surface wind fields. We will evaluate the ICON-simulated wind field distributions using reanalysis in a revised paper.*

2. Lack of detailed scientific explanations

The Result section focused on describing the obvious statistics from the presented figures without further digging into the science or physical mechanisms behind the "phenomenon". In general, lacking solid scientific explanation makes the presented paper more like an experimental report rather than an academic paper. Many open questions remain unexplained, including but not limiting to e.g., what could be the extra INP sources over sea ice? Does organic INPs representative enough in the Arctic environment? What are the driving forces behind more blowing snow/SIP over ice compared to the ocean? Why the contrast in ice crystal numbers different in different setup and temperature ranges? Etc. The storyline is completely unclear without addressing these questions arise from your results.

*The reviewer raises three questions. The first is on potential sources for INP over sea ice. We now provide a more thorough discussion of previous research on this point. Exact source strengths at a large scale of specific types are of course unknown (else it would not be an open science question). As such, our study is a sensitivity study: could this be an explanation for the sea ice – ocean difference in Ni? We make this now more clear in the revised manuscript. The second one is on blowing snow and SIP. Blowing snow of course can be lifted only from sea ice, not from the ocean. For SIP there is no evident difference, which we clarify now in the revision. It could just be due to differences in dynamic or microphysical conditions. Finally why contrast ice number in different temperature ranges? This is because the first-order determinant of Ni is the temperature and without stratification any detected signal may be confounded. We explain this now in the revision.*

3. Selective reporting of results

Even for the presented results, they are sometimes not accurately demonstrated, or the authors seem to manually pick the results that are more supportive or significant to explain their hypotheses. It was mentioned several times some results are not shown when they are not significant or against the hypotheses (e.g., lines 194, 294, and 309). In my opinion, it is unfair to conclude directly without showing the results because the readers should reserve the rights to make judgement out of an unbiased dataset provided by authors.

*We believe the manuscript is better readable if not every result that does not support the main conclusions is shown. But since the reviewer had the impression that instead, some results were hidden, we now provide all "not shown" results as Supplementary material.*

4. Insufficient and misapplied citations

As stressed above, in the Method and Result sections regarding ICON model setup, many configurations/parameters were announced without convincing reasonings or appropriate citations of relevant work. In addition, the author did not credit the proper citation of Arctic data, e.g., in lines 333-334, Petters and Wright (2015) was wrongly cited for comparing the Arctic INP data, which represents the global precipitation samples.

*In the revision, we now include more references on other studies which used the same or similar parameterizations to simulate Arctic clouds with the ICON model. We also clarify how we meant to cite the study by Petters and Wright (not focusing on the Arctic in this case).*

5. Missing Discussion Section

A Discussion section is missing, which should necessarily illustrate the limitation of the presented study, including e.g., the experimental setup (e.g., selection of scaling parameters, shortage of data due to very wide temperature bins); factors/hypotheses that are not examined but could have influence in the resulting difference in observed ice crystal numbers.

*In response to the reviewer remark, we now separate out a Discussion Section in which the discussion aspects are taken up in an extended way. We also include now a more elaborate discussion of the limitations of the present study.*

Regarding figures and data presentation, Figures 3–5 could be improved substantially:

• Consider using more effective color scales and symbols to make the data clearer.

*We hope the revised Figure is now more in line with the reviewer's expectations.*

• Eliminate redundant information or merge panels if they convey similar points; for instance, the right panels might be replaced with a more concise statistical significance test.

*While we agree with the reviewer that in principle the right panels are redundant with the left ones, it still seems useful to show them since it is the differences we are mostly interested in. In turn, one could consider omitting the left panels but then again it is useful for the reader to know the absolute numbers of both cases. In balance of these arguments, we would prefer to keep the figures as they are.*

• Narrowing the temperature bins or rearranging the experiments into a single figure could offer a clearer comparative perspective.

*We tried to collect all experiments in one figure, but this made the figure rather too crowded and not clear information could be drawn. Thus, we believe that the previous approach is the most pertinent one. Also, the rather wide temperature bins are designed as such in order to stick to the format of the first study (Papakonstantinou-Presvelou et al., 2022)*

Finally, the manuscript's language and structure would benefit from a thorough review, ideally by a native English speaker. Several informal expressions should be replaced with more precise academic language, unnecessary repetitions can be removed, and grammar/spelling errors need to be addressed. The paper would also benefit from overall conciseness, especially in the Introduction.

*We now thoroughly revised the language throughout the paper, aiming for more conciseness where possible. We will also ask Copernicus for the copy-editing help for a final version.*

I look forward to following up this interesting research topic and the improved version of this paper with more convincing scientific storyline and evidence, clearer and more concise demonstration of results, and more professional and precise language.

*We thank the reviewer for these final constructive comments.*